# Protocol for the challenge non-typhoidal *Salmonella* (CHANTS) study: a first-in-human, in-patient, double-blind, randomised, safety and dose-escalation controlled human infection model in the UK

Christopher Smith [iD],[1] Emma Smith,[1] Anna Rydlova,[1] Robert Varro,[1] Jay C D Hinton,[2] Melita A Gordon,[2,3] Robert K M Choy,[4] Xinxue Liu [iD],[5,6] Andrew J Pollard [iD],[5,6] Christopher Chiu,[1] Graham S Cooke,[1] Malick M Gibani [iD] [1]

For numbered affiliations see end of article.

**Correspondence to**
Dr Malick M Gibani;
m.gibani@imperial.ac.uk

## ABSTRACT

**Introduction** Invasive non-typhoidal *Salmonella* (iNTS) serovars are a major cause of community-acquired bloodstream infections in sub-Saharan Africa (SSA). In this setting, *Salmonella enterica* serovar Typhimurium accounts for two-thirds of infections and is associated with an estimated case fatality rate of 15%–20%. Several iNTS vaccine candidates are in early-stage assessment which—if found effective—would provide a valuable public health tool to reduce iNTS disease burden. The CHANTS study aims to develop a first-in-human *Salmonella* Typhimurium controlled human infection model, which can act as a platform for future vaccine evaluation, in addition to providing novel insights into iNTS disease pathogenesis.

**Methods and analysis** This double-blind, safety and dose-escalation study will randomise 40–80 healthy UK participants aged 18–50 to receive oral challenge with one of two strains of *S*. Typhimurium belonging to the ST19 (strain 4/74) or ST313 (strain D23580) lineages. 4/74 is a global strain often associated with diarrhoeal illness predominantly in high-income settings, while D23580 is an archetypal strain representing invasive disease-causing isolates found in SSA. The primary objective is to determine the minimum infectious dose (colony-forming unit) required for 60%–75% of participants to develop clinical or microbiological features of systemic salmonellosis. Secondary endpoints are to describe and compare the clinical, microbiological and immunological responses following challenge. Dose escalation or de-escalation will be undertaken by continual-reassessment methodology and limited within prespecified safety thresholds. Exploratory objectives are to describe mechanisms of iNTS virulence, identify putative immune correlates of protection and describe host–pathogen interactions in response to infection.

**Ethics and dissemination** Ethical approval has been obtained from the NHS Health Research Authority (London—Fulham Research Ethics Committee 21/PR/0051; IRAS Project ID 301659). The study findings will be disseminated in international peer-reviewed journals and presented at national/international stakeholder meetings. Study outcome summaries will be provided to both funders and participants.

**Trial registration number** NCT05870150

## STRENGTHS AND LIMITATIONS OF THIS STUDY

⇒ This first-in-human non-typhoidal *Salmonella* (NTS) controlled human infection model uses well-characterised *S*. Typhimurium strains (ST19 and ST313), which are representative of currently circulating isolates accountable for both diarrhoeal and invasive disease phenotypes.

⇒ A double-blind, randomised study design permits objective evaluation of differential clinical phenotypes induced by bacterial challenge.

⇒ The continual reassessment method for dose-finding, compared with rule-based design, improves trial efficiency and increases the probability of finding the correct challenge dose.

⇒ This study is limited by recruitment of healthy UK volunteers, whose responses to challenge will differ from children and immunocompromised adults most commonly affected by invasive NTS (iNTS) in sub-Saharan Africa.

## INTRODUCTION

Non-typhoidal *Salmonella* (NTS) infections are most commonly associated with a self-limiting diarrhoeal illness (dNTS).[1] In contrast, invasive NTS disease (iNTS) typically presents as a non-specific febrile illness—often without diarrhoea—following invasion into sterile sites causing bloodstream infection (BSI), meningitis or sepsis.[2] iNTS is particularly common in sub-Saharan Africa (SSA) where most recent estimates reported 535 000 cases,

77 500 deaths and 4.26 million lost Disability adjusted life years (DALYs) in 2017.[3] Case fatality rates are estimated to range between 15% and 20%. The high case fatality reflects that invasive disease occurs almost exclusively in individuals with impaired immunity. Recognised host risk factors include malnutrition, malaria and chronic anaemia in children, and HIV infection and malignancy in adults.[2 4] Effective treatment is also increasingly challenging, highlighted by the WHO's inclusion of *Salmonella enterica* as one of the priority pathogens posing the greatest risk to human health through expanding antimicrobial resistance (AMR).[5]

iNTS serovars isolated in community-acquired BSI exhibit geographical variation. In Africa, *Salmonella enterica* subspecies *enterica* serovars Typhimurium and Enteritidis (henceforth S. Typhimurium and S. Enteritidis) are responsible for >90% of all iNTS BSI, with S. Typhimurium accounting for two-thirds of this burden.[6–8] Sequencing of isolates from across Africa has identified S. Typhimurium multilocus sequence type (ST) 313 as the major cause of invasive disease, which is distinct from STs 19 and 34 that are more frequently associated with diarrhoeal disease (dNTS) in high-income settings such as the UK. ST313 has been shown to have undergone sequential evolutionary changes that converge towards a genotype associated with invasive disease.[9]

To date, there are no licensed iNTS vaccines available for use in humans. Development of effective vaccines is urgently required and may contribute to reduced case numbers, reduced antibiotic use and prevention of AMR.[10] Several candidate vaccines are under evaluation, including in bivalent or multivalent formulations, which could target both typhoidal and non-typhoidal *Salmonella* serovars.[11] Vaccine development for iNTS is hampered—in part—by an incomplete understanding of mechanisms and determinants of immunity during natural infection. Investigation into these mechanisms could support iNTS vaccine development programmes through identification of immune correlates of protection. In addition, vaccine development is hindered by the substantial time and financial commitments required to conduct large-scale phase III trials in field settings to obtain efficacy data.

Controlled human infection models (CHIMs) have a strong track record in accelerating vaccine development, particularly for enteric pathogens.[12 13] For example, the live attenuated cholera vaccine CVD 103-HgR was licensed as a travel vaccine on the basis of safety, immunogenicity and efficacy data arising from CHIM studies.[14] More recently, a programme of typhoidal *Salmonella* CHIM studies demonstrated efficacy of a Vi-tetanus toxoid (Vi-TT) conjugate vaccine against *S. typhi* (Typbar-TCV, Bharat Biotech).[15] These data contributed to recommendations from WHO SAGE to support programmatic use of Vi-TT in high-burden settings and were supported by subsequent phase III trials demonstrating similar efficacy in Nepal, Bangladesh and Malawi.[16–18]

The CHANTS study has been funded by the Wellcome Trust to establish a first-in-human NTS CHIM, with the aim of addressing key knowledge gaps in dNTS and iNTS disease pathogenesis and immunity. We will conduct a double-blind, randomised, safety and dose-escalation study with the primary objective of determining the infectious dose required for 60%–75% of participants to develop systemic salmonellosis. Successfully establishing a reproducible NTS CHIM could provide valuable insights into mechanisms of protective immunity and serve as a platform for future evaluation of candidate iNTS vaccines and therapeutics.

## METHODS AND ANALYSIS
### Study design
The CHANTS study is a double-blinded, randomised, quarantine-based, CHIM, which aims to establish the infectious dose required for healthy participants to develop salmonellosis following oral challenge with either *S.* Typhimurium ST313 (strain D23580) or ST19 (strain 4/74). A predicted total of 40–80 healthy adult volunteers who are eligible, and consenting, will be randomised 1:1 to receive oral challenge with either D23580 or 4/74. The study will be conducted at Imperial College London with clinical support from partner tertiary hospitals at Imperial College Healthcare NHS Trust. Participant screening and recruitment commenced in August 2023, and the study is anticipated to conclude in Summer 2025.

### Patient and public involvement
Patient and public engagement activities have been conducted throughout the protocol design stage, gaining valuable insights from the involvement of key stakeholders. Focus groups have been conducted with support of the NIHR Imperial Biomedical Research Centre patient experience research group. Discussions with members of the public in this forum—including past participants of previous CHIMs—have identified key considerations for the safe and acceptable conduct of this trial and supported the development of clear and thorough participant-facing information to underpin informed consent. Future similar activities are planned in regions with high iNTS burden to establish acceptability of potential future transfer of an iNTS CHIM to settings most affected by the disease under investigation.

### Study objectives and outcomes
The primary objective is to determine the minimum infectious dose in colony-forming units (CFU) required for 60%–75% of healthy participants to develop clinical or microbiological features of systemic salmonellosis following oral challenge with either *S.* Typhimurium D23580 or 4/74. Systemic salmonellosis is defined as the development of either: (1) fever ≥38°C on ≥2 occasions ≥12 hours apart or (2) *S.* Typhimurium bacteraemia at any time point. As outlined above, exposure to *S.* Typhimurium can result in heterogeneous disease phenotypes, ranging from asymptomatic gastrointestinal colonisation to BSI. Secondary endpoints include determining the

**Table 1** Primary and secondary study objectives and outcomes

| | Objective(s) | Outcome(s)/endpoint(s) |
|---|---|---|
| Primary | Determine the minimum infectious dose (CFU) of *Salmonella* Typhimurium required for 60%–75% of volunteers to develop systemic Salmonellosis following oral challenge. | The proportion of participants who develop:<br>1. Fever ≥38°C on ≥2 occasions ≥12 hours apart.<br>2. *S.* Typhimurium bacteraemia at any time point. |
| Secondary | Describe colonisation rates following oral challenge at different doses. | The proportion from whom *S.* Typhimurium is isolated from stool on ≥2 occasions ≥48 hours from challenge at different doses of each strain. |
| | Describe gastroenteritis rates following oral challenge at different doses. | The proportion of participants at different doses of each strain developing:<br>1. Severe diarrhoea.*<br>2. Moderate diarrhoea† plus fever ≥38°C on ≥1 occasion and/or ≥1 grade 2 gastrointestinal symptoms (abdominal pain, nausea, vomiting, tenesmus). |
| | Determine persistent fever rates following oral challenge at different doses. | The proportion of participants who develop fever ≥38°C on ≥2 occasions ≥12 hours apart at different doses of each strain. |
| | Describe bacteraemia rate following oral challenge at different doses. | The proportion of participants developing *S.* Typhimurium bacteraemia at any time point following challenge at different doses of each strain. |
| | Describe the rates of systemic *Salmonellosis* according to an alternative composite diagnostic criterion following oral challenge at different doses. | The proportion of participants meeting the criteria for a composite diagnosis of *Salmonellosis* at different doses of each strain defined as any of:<br>1. *S.* Typhimurium isolated from stool on ≥2 occasions ≥48 hours from challenge.<br>2. Gastroenteritis.<br>3. Fever ≥38°C on ≥2 occasions ≥12 hours apart.<br>4. *S.* Typhimurium bacteraemia. |
| | Describe the safety of oral challenge with *S.* Typhimurium 4/74 and D23580 strains. | The proportion of participants at different doses of each strain reporting: adverse events, adverse events of special interest, SAEs, SUSARs, concomitant medication usage. |
| | Compare clinical features following oral challenge with *S.* Typhimurium strains 4/74 or D23580. | A comparison of the clinical features of *Salmonella* infection after challenge, with specific reference to the proportion in each group developing:<br>1. Diarrhoea (volume, frequency, consistency, gastrointestinal symptom severity score, time to onset).<br>2. Any fever ≥38°C (including fever clearance time).<br>3. Solicited symptoms at any time: headache, malaise, anorexia, abdominal pain, nausea, vomiting, dysentery, myalgia, arthralgia, cough, rash (including measurement of severity and duration from onset to resolution). |
| | Compare microbiological features of gastrointestinal infection following oral challenge with *S.* Typhimurium 4/74 or D23580. | Gastrointestinal *Salmonella* infection after challenge with 4/74 or D23580 strains, with specific reference to:<br>1. Time to onset in days in each group from challenge to first stool sample positive for *S.* Typhimurium by culture and/or PCR.<br>2. Duration of stool shedding.<br>3. The magnitude of stool shedding measured in CFU/mL in quantitative stool culture analysis. |
| | Compare microbiological features of bloodstream infection following oral challenge with *S.* Typhimurium 4/74 or *S.* Typhimurium D23580 strains | *Salmonella* bloodstream infection after challenge, with specific reference to:<br>1. The proportion of participants in each group developing any bacteraemia.<br>2. The severity of bacteraemia in each group as measured by time to onset, duration, clearance time, magnitude quantified in CFU/mL. |
| | Compare biochemical and haematological laboratory parameters following oral challenge. | Absolute values of laboratory parameters from time of challenge to day 28 and/or day 90, with specific reference to: haemoglobin, white cell count and differential, urea and electrolytes, liver function tests. |

Continued

**Table 1** Continued

| | Objective(s) | Outcome(s)/endpoint(s) |
|---|---|---|
| | Describe and compare the serum antibody response following oral challenge with *S.* Typhimurium 4/74 or D23580 strains. | Serum samples measured at baseline and post challenge time points:<br>1. *S.* Typhimurium O-specific polysaccharide serum IgG and IgA concentration measured by ELISA.<br>2. *S.* Typhimurium-specific serum IgG and IgA concentration against other *S.* Typhimurium antigens measured by ELISA.<br>3. Serum bactericidal antibody titres against *S.* Typhimurium<br>4. *S.* Typhimurium-specific antibody secreting cell and memory B-cell responses measured by ELISPOT<br>5. Other functional antibody activity measurements including systems serology platforms. |
| | Describe and compare the mucosal antibody response following oral challenge with *S.* Typhimurium 4/74 or D23580 strains. | Saliva and/or stool samples measured at baseline and post challenge time points:<br>1. *S.* Typhimurium-specific IgG and IgA concentration measured by ELISA.<br>2. Bactericidal antibody titres.<br>3. Other functional antibody activity measurements including systems serology platforms. |
| | Describe and compare cell-mediated immune response following oral challenge with *S.* Typhimurium 4/74 D23580 strains. | Absolute values and intergroup comparison of laboratory assays performed on peripheral blood mononuclear cells measured at baseline and post challenge time points:<br>1. Lymphocyte populations as measured by flow cytometry and/or CyTOF.<br>2. *S.* Typhimurium-specific cell-mediated immune responses as measured by ELISPOT and flow cytometry and/or CyTOF. |
| | To describe participant experience following oral challenge with *S.* Typhimurium 4/74 or D23580 strains | Descriptive statistics following administration of a structured questionnaire at baseline and postchallenge. |

*Severe gastroenteritis: ≥6 loose/liquid stools (Bristol types 6–7) and/or >800 g of loose/liquid stools in a rolling 24-hour period and/or ≥2 stools with gross blood in 24 hours.
†Moderate gastroenteritis: 4–5 loose/liquid stools (Bristol types 6–7) and/or 400–800 g in a rolling 24-hour period.
CFU, colony-forming unit; CyTOF, Cytometry by time of flight; SAE, Serious adverse events; SUSAR, Suspected Unexpected Serious Adverse Reaction.

proportion of participants who develop outcomes along this disease spectrum, including *Salmonella* colonisation, diarrhoeal disease (dNTS), systemic symptoms or severe salmonellosis. Secondary study outcomes also aim to describe the immune response to *S.* Typhimurium infection (table 1).

In this first-in-human study, we have a unique opportunity to elucidate mechanisms of bacterial virulence in diarrhoeal disease secondary to NTS (dNTS) and iNTS, identify immune correlates of protection, describe host–pathogen interactions, changes in host microbiota in response to infection and develop novel diagnostic platforms (online supplemental table 1).

### Recruitment and eligibility
Several strategies will be employed to recruit study participants, including advertising throughout local hospitals, GP surgeries and higher education institutions; media and website advertising; direct email to members of the NIHR Imperial Clinical Research Facility healthy volunteer database; direct mail-out and email to potentially eligible participants identified via National Health Applications and Infrastructure Services. Consenting healthy adult males and non-pregnant females aged 18–50

without pre-existing risk factors for severe iNTS or chronic infection will be recruited. Strict eligibility criteria have been designed to mitigate risks of severe disease, chronic carriage or development of focal seeded infection. Detailed screening will include comprehensive medical history, physical examination, baseline blood counts, biochemical profiles, screening for immunosuppression and abdominal ultrasonography to exclude subclinical biliary or abdominal aortic disease (see online supplemental table 2 for full eligibility criteria). Individuals who have had documented prior *Salmonella* infection, or received live-attenuated oral Ty21a typhoid vaccine, will be excluded due to potential cross-protective immune responses against *S.* Typhimurium. Further temporary exclusion criteria will be reviewed prior to oral challenge to exclude intercurrent infection or antimicrobial use which may bias results (online supplemental table 3).

### Study interventions
Consenting participants who fulfil eligibility criteria will enter the study at day −7 (7 days prior to oral bacterial challenge). Days 0–7 will take place under clinical observation in a quarantine research facility at Imperial College Healthcare NHS Trust, with subsequent daily outpatient

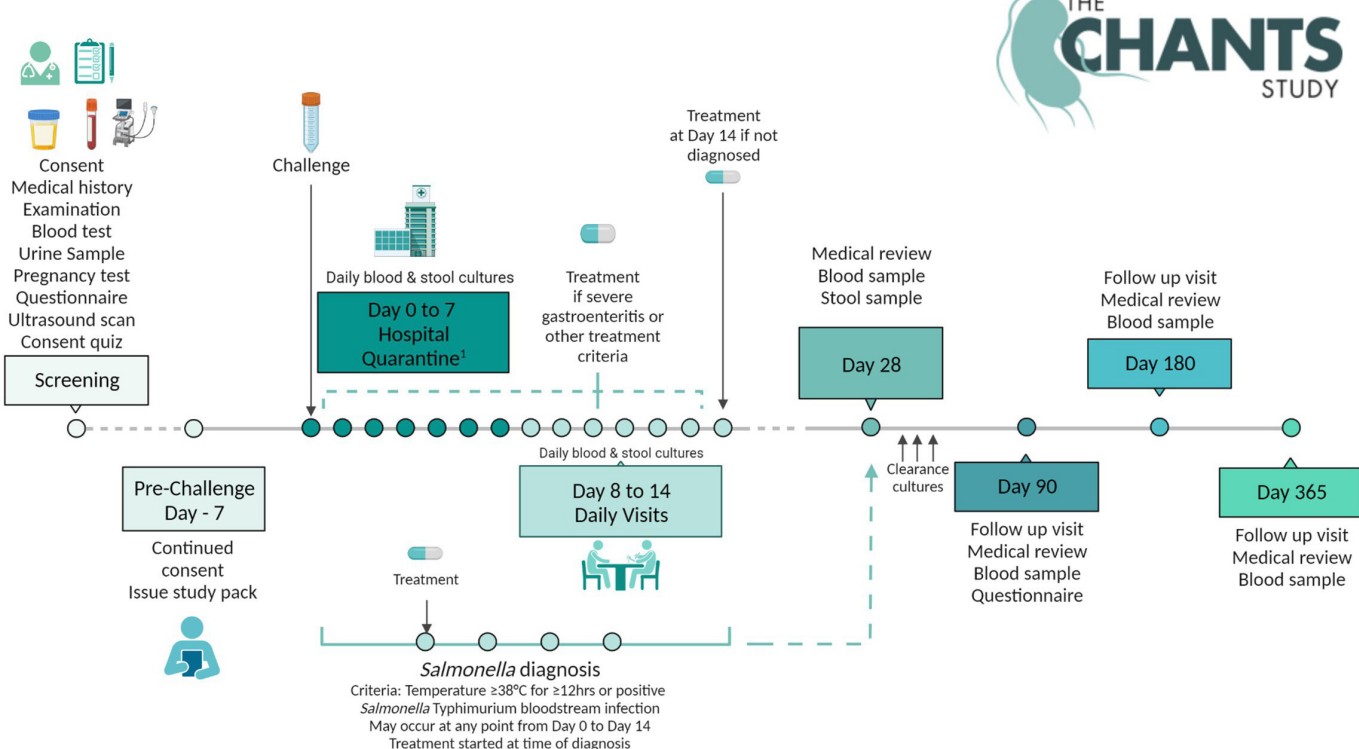

**Figure 1** Overview of the participant journey from screening to study completion at 1 year of follow-up.

reviews on days 8–14, prior to long-term follow-up visits at days 28, 90, 180 and 365. The overall participant journey is illustrated in figure 1.

## Challenge strain selection and manufacture

Challenge strains have been selected to maximise future utility of the model in supporting iNTS vaccine development. Wild-type ST313 (D23580) and ST19 (4/74) are phylogenetically representative of contemporary disease-causing isolates, have a traceable history from isolation and have undergone detailed genotypic, phenotypic and transcriptomic characterisation.[19 20] Bacterial stocks have been manufactured to GMP standard by the Walter Reed Army Institute of Research, Maryland, USA, in collaboration with PATH (Seattle, USA), with strains originally drawn from the UK health security agency (UKHSA) National Collection of Type Cultures.

ST313 lineage 2 reference strain D23580 was originally isolated from a blood culture obtained from a 24-month-old HIV-negative child at Queen Elizabeth Central Hospital, Blantyre, Malawi and is considered an archetypal isolate representative of strains causing iNTS in SSA. ST19 (4/74), originally isolated from a calf in England in 1974, is considered an archetypal strain causative of dNTS and representative of current clinical infections. D23580 harbours a distinct prophage and plasmid repertoire compared with 4/74, including a virulence plasmid pSLT encoding resistance to ampicillin, chloramphenicol, streptomycin, sulfonamides and trimethoprim.[20] Both strains are susceptible to ciprofloxacin, azithromycin and third-generation cephalosporins,

ensuring a range of treatment options are available for challenged participants. The rationale for strains selected for use in this model, and their relative merits, have been discussed in detail elsewhere.[21] We hypothesise that ST313 (D23580) will be associated with an invasive phenotype compared with the enterocolitis anticipated as a result of challenge with ST19 (4/74).

## Randomisation and blinding

Block randomisation (block size=10) to strain allocation will be carried out on or after day −7. An initial sentinel cohort (n=10) of will be randomised 1:1 (block sizes=2, 4, 4) to receive a starting dose $1–5×10^3$ CFU of either strain 4/74 or D23580. This starting dose has been determined based on prior experience from typhoidal CHIMs and continual reassessment method (CRM) dose prediction simulations (see the Statistical analysis section).[15 22–24] Unblinded attack rates and safety data will be reviewed by the data safety and monitoring committee (DSMC) prior to further participant randomisation (see the Governance section). Randomisation lists will be generated and stored on a secure server accessible only by unblinded study team members. All clinical staff and participants will remain blinded to challenge allocation for the duration of the study.

## Dose escalation/de-escalation

Dose escalation and de-escalation decisions will be undertaken by the CRM. Escalation for both strains will occur in parallel, starting at $1–5×10^3$ CFU, increasing to a maximum of $1–5×10^6$ CFU or decreasing down to $1–5×10^1$

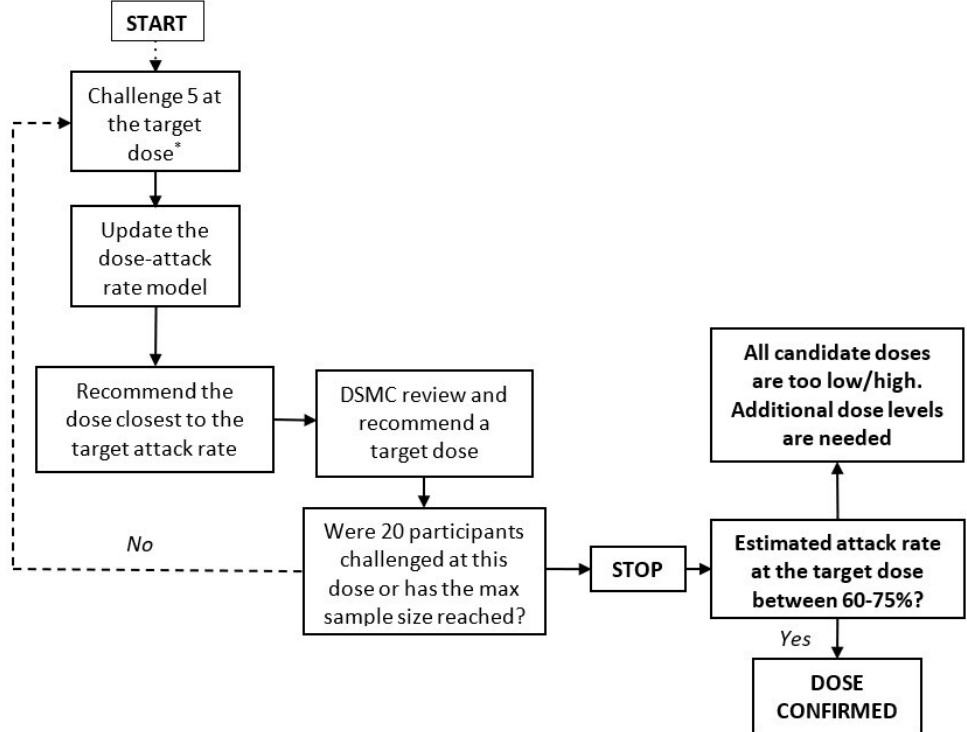

**Figure 2** Decision making algorithm for dose escalation/de-escalation. Starting at 1-5 x10e3 CFU to reach the primary endpoint. *The target dose for the first cohort will be 1-5 x 10e3 CFU, and the first cohort of 5 participants will be divided into 3 groups (1, 2, and 2 participants) and challenged with an interval of at least 7 days. For the rest of the cohorts, they will be challenged on the same day, if possible. DSMC, data safety and monitoring committee.

CFU. Participants will be randomised to receive one of the two challenge agents at the dose determined by the CRM until target dosing is established for one agent. From that point, the remaining participants will be challenged with the other agent. We aim to recruit between 40 and 80 participants in total (20–40 for each challenge agent). The final challenge dose will be defined as one that achieves a 60%–75% attack rate. The DSMC will be provided with an updated report after each cohort of five completes challenge. The dose escalation/de-escalation decision algorithm is illustrated in figure 2.

### Challenge procedures
Participants will receive oral bacterial challenge with either D23580 or 4/74 on study day 0, following admission to an isolation side room within the quarantine facility. Assessments will be conducted to confirm the absence of any temporary exclusion criteria (see online supplemental table 3) prior to challenge. Female participants will also undergo pregnancy testing both at screening and immediately prior to challenge. Both S. Typhimurium challenge agents used in this study will be administered via the oral route with sodium bicarbonate buffer to achieve gastric acid neutralisation. Participants will fast for 90 min before and after challenge. The solution for ingestion containing either wild type S. Typhimurium 4/74 or D23580 will be prepared in a class II biological safety cabinet that is solely used for the purpose of preparing the challenge solution.

### Clinical assessment during quarantine
Participants will be reviewed daily during admission to the quarantine facility by a clinical member of the study team. Extensive assessments will be undertaken, including physical examination, recording vital signs, assessment of mood and review of participant symptom diaries. Blood, stool and saliva specimen collection will be undertaken according to timelines outlined in online supplemental table 4. Supportive therapy will be provided as required for symptom control, for example, oral rehydration solution or intravenous fluids, analgesia and antiemetics.

### Infection prevention and control
Participants will be managed using infection control protocols appropriate for enteric infections. Study investigators will adhere to strict hand-hygiene policies of the research facility and use enteric personal protective equipment for all study procedures. Participants will receive clear instruction on hand-hygiene procedures—including illustrated guidance—to be adhered to during and after quarantine, in addition to guidance on cleaning protocols for bathrooms in the home environment following discharge. Participants will be nursed within single-occupancy side rooms with ensuite toilet facilities for the duration of quarantine. The near-patient environment, including door handles, taps, toilet flushes and bed rails, will be decontaminated daily throughout the quarantine period in accordance with local policies. Assessment of *Salmonella* environmental contamination

within the treatment facility will be undertaken as an exploratory study objective, using culture and molecular methods to detect the presence of *S.* Typhimurium in the participant's immediate surroundings and in high-traffic clinical areas.

Close contacts of participants, including household contacts, will be offered the opportunity to be screened for *Salmonella* infection, which will involve obtaining three stool samples 48 hours apart a minimum of 7 days after the participant has begun antibiotic treatment. If either stool culture of a household contact is positive, he/she will be referred to a consultant in infectious diseases for appropriate antibiotic management and the local health protection unit will be informed.

### Antimicrobial therapy

Treatment of *Salmonella* infection in healthy immunocompetent adults poses a number of specific challenges. Antimicrobial therapy is indicated for invasive disease but may prolong shedding in patients with paucisymptomatic gastroenteritis. We have elected to offer rescue treatment in specific circumstances. The rationale for this decision is discussed elsewhere.[21] Antibiotic therapy will be initiated in all participants who fulfil criteria for *Salmonella* diagnosis through reaching either or both of the defined primary endpoints of persistent fever or BSI. Treatment will also be commenced in participants who do not achieve the primary outcome measures but reach any of the other prespecified treatment criteria outlined in box 1. Treatment for BSI will comprise oral ciprofloxacin 500 mg two times per day for 10–14 days, or second-line therapy of intravenous ceftriaxone 1 g once daily for up to 14 days. All other treatment indications (systemic illness, gastroenteritis, asymptomatic/paucisymptomatic shedding) will be treated with oral ciprofloxacin 500 mg two times per day for 5 days, or with second-line oral azithromycin 500 mg once daily for 5 days. Antibiotic treatment will not be routinely offered to participants who do not fulfil any prespecified treatment criteria. The full antibiotic treatment decision algorithm is illustrated in online supplemental figure 1.

### Discharge from quarantine

The duration of quarantine will be 7 days from the time of oral challenge (168 hours). Participants will be eligible for discharge if they fulfil the criteria outlined in box 2. Some participants may be discharged earlier than day 7 if they have been started on antibiotic therapy and completed 96 hours of follow-up after starting treatment. If these criteria are not met on day 7, then participants will be asked to remain in quarantine until discharge criteria are met. The discharge decision-making algorithm is illustrated in online supplemental figure 2.

Discharge criteria have been designed to maximise participant clinical safety in addition to minimising risk of secondary onward transmission. Current UKHSA guidance does not mandate the collection of clearance stool cultures following an episode of gastroenteritis secondary

---

**Box 1  Criteria for commencing antibiotic therapy following oral bacterial challenge**

**Antibiotics are commenced if any of the following apply.**
Any participant with *Salmonella* Typhimurium bacteraemia.
Fever ≥38°C for ≥12 hours.
Any participant with severe gastroenteritis*.
Moderate gastroenteritis** plus:
⇒ Fever ≥38°C on one occasion.
⇒ ≥1 grade 2*** systemic symptoms.
Any participant with three or more of the following symptoms on the same day at grade 2 or higher.
⇒ Headache.
⇒ Fatigue/malaise.
⇒ Anorexia.
⇒ Abdominal pain.
⇒ Nausea.
⇒ Vomiting.
⇒ Myalgia.
⇒ Arthralgia.
⇒ Cough.
Any participant from whom *Salmonella* has been detected from at least two stool culture/PCR and 24 hours apart who has not received antibiotics by day 14 postchallenge.
Any participant in whom antibiotic use is felt to be clinically necessary (as decided by a medically qualified study doctor).

*Severe gastroenteritis: ≥6 loose/liquid stools (Bristol types 6–7) and/or >800 g of loose/liquid stools in a rolling 24-hour period and/or ≥2 stools with gross blood in 24 hours.
**Moderate gastroenteritis: 4–5 loose/liquid stools (Bristol types 6–7) and/or 400–800 g in a rolling 24-hour period.
***Grade 0=not present; grade 1=present but no interference with activity; grade 2=some interference with activity; grade 3=significant; prevents daily activity; grade 4: Emergency department (ED) visit or hospitalisation.

---

to NTS infection. Instead, the absence of diarrhoea for at least 48 hours is considered sufficient for release from isolation.[25] In this study, clearance stool cultures will be obtained to meet a secondary study objective of measuring predicted differences in patterns of shedding between 4/74 and D23580 strains.

### Governance

An independent data and safety monitoring committee (DSMC) has been appointed to provide real-time oversight of safety and trial conduct. The DSMC will review unblinded safety data throughout the study according to a predefined DSMC charter. The committee will have access to data and, if required, will monitor these data, and make recommendations to the study investigators on whether there are any ethical or safety reasons the trial should not continue. The DSMC will review the attack rate and advise on dose escalation/de-escalation decisions. Study monitoring will also be undertaken by a Contract Research Organisation, separate to and independent from the DSMC, to ensure safety and compliance with the approved study protocol. Strict data management plans

---

**Box 2   Quarantine discharge criteria**

**Participants will be discharged from the inpatient quarantine unit if the following criteria are met:**
⇒ Medically fit for discharge in the opinion of the study physician.
⇒ Complete resolution of diarrhoea (Bristol stool types 6–7) for 48 hours.
⇒ Seven days (168 hours) have elapsed since challenge.

**For patients diagnosed with Salmonellosis from day 0 to day 7, the following criteria apply:**
⇒ Antibiotic treatment has been initiated and patient has completed 96 hours follow-up.
⇒ Resolution of *Salmonella* Typhimurium bacteraemia (if applicable).

---

will be adhered to with compliance to Good Clinical Practice guidelines.

## Safety

As with all CHIM, maintenance of participant safety during and after the study is paramount. In this model, general risks associated with participation may include study fatigue, repeated phlebotomy procedures and consequences of isolation in quarantine. Risks more directly attributable to challenge agents include symptomatic infection, complications of antibiotic treatment (in the case of bacterial challenge), and the small risk of long-term sequelae. Potential risks have been given extensive consideration, with risk mitigation strategies devised to minimise possible impact on participants. The potential complications of participation—and risk mitigation strategies—have been discussed elsewhere.[21] In addition to participant safety, careful consideration has been given to ensure safety of study investigators and close contacts. Clear infection control protocols have been developed (as outlined above), in addition to close liaison with UKHSA to monitor and address any potential secondary transmission events, including within households following discharge from the quarantine facility.

## Statistical analysis

This dose-finding study will use the CRM to determine the target dose that can achieve a 60%–75% attack rate.[26 27] To our knowledge, this will be the first dose-finding CHIM to employ CRM methodology. The advantages over traditional rule-based dose-finding studies are that the CRM method borrows information across all dose levels, which increases likelihood of finding the true dose for a given sample size compared with rule-based methods in the majority of scenarios, particularly when the maximum sample size is small. There is no formal sample size calculation for this study. The maximum sample size will be 40 participants for each challenge strain. The operating characteristics of the CRM and the rule-based approach are illustrated in online supplemental table 5. An International Committee of Medical Journal Editors (ICMJE) data availability statement is included in online supplemental table 6, applicable after conclusion of the study.

## Limitations

The primary limitations of the CHANTS study—shared with most CHIM—stem from undertaking the study in a healthy adult volunteer group which may not accurately reflect natural disease acquisition and pathogenesis. There are clear host factor differences between this group and individuals most susceptible to iNTS disease. However, as with prior CHIMs undertaken in healthy participants, we anticipate this infection model will generate key insights into *Salmonella* infection biology with future translational applications. Successful development of an *S.* Typhimurium CHIM, with an established safety profile and infectious dose, could be transferred to an endemic setting for future investigation in a population more closely resembling those at greatest risk of disease, as has been achieved previously in a range of enteric and non-enteric pathogen challenge models.[28–32]

In addition to inherent differences between our study population and those at greatest risk of iNTS, there are limitations in achievable study endpoints. Our primary endpoint of persistent fever, with or without BSI, does not necessarily reflect the severe clinical phenotypes an iNTS vaccine would aim to protect against in field settings. However, in view of the heterogeneous clinical phenotypes associated with NTS infection, ranging from asymptomatic colonisation to septic shock, this endpoint is felt to represent the optimal trade-off between clinical manifestations of disease and patient safety within the context of a CHIM. The relative merits of endpoint selection and their rationale have been discussed previously.[21]

## ETHICS AND DISSEMINATION

This study protocol has been reviewed and approved by the NHS Health Research Authority (London—Fulham Research Ethics Committee 21/PR/0051; IRAS 301659) and is registered on ClinicalTrials.gov (registration number NCT05870150). The sponsor is Imperial College London, and the study is indemnified under institutional and NHS insurance policies. Our findings will be of significant international interest, with several potential future applications for the assessment of iNTS vaccines and novel therapeutics. We will publish all findings in peer-reviewed journals, and ensure presentation at national and international conferences, in addition to close liaison with key stakeholders, participants and funders.

**Author affiliations**
[1]Department of Infectious Disease, Imperial College London, London, UK
[2]Institute of Infection, Veterinary and Ecological Sciences, University of Liverpool, Liverpool, UK
[3]Malawi Liverpool Wellcome Trust Clinical Research Programme, Kamuzu University of Health Sciences, Blantyre, Southern Region, Malawi
[4]PATH, Seattle, Washington, USA
[5]Oxford Vaccine Group, Department of Paediatrics, Oxford University, Oxford, UK
[6]NIHR Oxford Biomedical Research Centre, Oxford, UK

**Contributors** Conceptualisation and study design: CS, ES, AR, RV, JCDH, MAG, RKMC, XL, AJP, CC, GSC and MMG. Data analysis plan: CS, XL, GSC and MMG.

Drafting of protocol: CS, ES, AR, RV, GSC and MMG. Protocol amendments: CS, RKMC, AJP, XL and MMG.

**Funding** The CHANTS study is funded by The Wellcome Trust (grant number 224029/Z/21/Z: awarded to principal investigator MMG). CS, GSC, CC and MMG are supported, in part, by the National Institute of Health Research Imperial Biomedical Research Centre.

**Competing interests** None declared.

**Patient and public involvement** Patients and/or the public were involved in the design, or conduct, or reporting, or dissemination plans of this research. Refer to the Methods section for further details.

**Patient consent for publication** Not applicable.

**Ethics approval** London (Fulham) REC reference: 21/PR/0051

**Provenance and peer review** Not commissioned; externally peer reviewed.

**ORCID iDs**
Christopher Smith http://orcid.org/0000-0001-7369-2034
Xinxue Liu http://orcid.org/0000-0003-1107-0365
Andrew J Pollard http://orcid.org/0000-0001-7361-719X
Malick M Gibani http://orcid.org/0000-0003-1781-0053

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
