## [Reviewer comments · BMJ Open]

ARTICLE DETAILS

TITLE (PROVISIONAL)	Protocol for the Challenge Nontyphoidal Salmonella (CHANTS) Study: a first-in-human, in-patient, double-blind, randomised, safety and dose-escalation controlled human infection model in the United Kingdom
AUTHORS	Smith, Christopher; Smith, Emma; Rydlova, Anna; Varro, Robert; Hinton, Jay; Gordon, Melita; Choy, Robert; Liu, Xinxue; Pollard, Andrew; Chiu, Christopher; Cooke, Graham; Gibani, Malick

VERSION 1 – REVIEW

REVIEWER	Lacey, Charles Jn University of York
REVIEW RETURNED	13-Nov-2023

GENERAL COMMENTS	The manuscript bmjopen-2023-076477, by Smith et al., describes the plans of CHANTS study (Challenge Nontyphoidal Salmonella – CHANTS). This is the plan to develop a CHIM for NTS aiming to gain insights into its physiopathology and to develop a tool for vaccine testing. Nontyphoidal salmonellosis is a severe disease with clinical presentation ranging from asymptomatic to sepsis with a high mortality rate. The development of the CHIM is well justified because of the social impact of NTS infection. In addition, the model will serve not only to accelerate an urgently needed vaccine, but also help the scientific community to understand the disease progression and presentation, perhaps allowing the identification of resistance surrogate markers. The protocol is well described, and it is clear a lot of effort has been employed to identify and mitigate risks to guarantee the safety of study participant, to plan accurate decision taking along the protocol, and to promote accurate and relevant data collection. The authors will use a novel approach to escalate/de-escalate pathogen dosing, based on real-time data analysis from all the doses used (CRM analysis). The aims are very clear and allow the best protocol progression to fulfill it. The protocol has received ethical approval and is expected to contribute significantly to the specific field of research. There are no results presented as the manuscript is a protocol description.
---

REVIEWER	Santiago, Helton C Universidade Federal de Minas Gerais
REVIEW RETURNED	13-Nov-2023

GENERAL COMMENTS	The manuscript bmjopen-2023-076477, by Smith et al., describes the plans of CHANTS study (Challenge Nontyphoidal Salmonella – CHANTS). This is the plan to develop a CHIM for NTS aiming to gain insights into its physiopathology and to develop a tool for vaccine testing. Nontyphoidal salmonellosis is a severe disease with clinical presentation ranging from asymptomatic to sepsis with a high mortality rate. The development of the CHIM is well justified because of the social impact of NTS infection. In addition, the model will serve not only to accelerate an urgently needed vaccine, but also help the scientific community to understand the disease progression and presentation, perhaps allowing the identification of resistance surrogate markers. The protocol is well described, and it is clear a lot of effort has been employed to identify and mitigate risks to guarantee the safety of study participant, to plan accurate decision taking along the protocol, and to promote accurate and relevant data collection. The authors will use a novel approach to escalate/de-escalate pathogen dosing, based on real-time data analysis from all the doses used (CRM analysis). The aims are very clear and allow the best protocol progression to fulfill it. The protocol has received ethical approval and is expected to contribute significantly to the specific field of research. There are no results presented as the manuscript is a protocol description.
---

VERSION 1 – AUTHOR RESPONSE

Reviewer: 1 (Dr. Charles Jn Lacey, University of York)

Very good paper and protocol. Just a few minor queries as below. Could the authors respond, and make any changes deemed necessary? An exclusion criterion is “Female participants who are pregnant, lactating or who are unwilling to ensure that they or their partner use effective contraception 30 days prior to challenge and continue to do so until two negative stool samples, ...” What is the definition of ‘effective contraception’? The investigators would presumably wish to avoid enrolling subjects who are using less effective methods (e.g. barrier methods), so an inclusive list of acceptable methods would be valuable.

We thank the reviewer for raising these important points. An inclusive list of acceptable methods of contraception has been added to the eligibility criteria supplementary table 1 with a qualifying statement: ‘All forms of contraception are considered effective with the exception of barrier methods and natural family planning. Accepted methods include hormonal contraceptive pills, intrauterine contraceptive devices/systems, long-acting injectable hormonal contraception, and contraceptive implants.’

Should female participants have pregnancy testing at screening and entry?

Thank you. As suggested, pregnancy testing for female participants takes place at screening and on admission to quarantine immediately prior to oral bacterial challenge. This has been included in the main text of the manuscript to highlight this practice (Line 274-275, Marked Copy).

An exclusion criterion is “Close household contact with young children (defined as those attending pre-school groups, nursery or those aged less than 2 years). I find that definition of ‘young children’ slightly confusing. It seems that if one’s child was 4 years old and attending nursery the parent would be excluded, whereas if the child was still at home not attending nursery the parent could be included. Would not a simple age-based criterion be better and safer?

Thank you for raising this important point. We have altered the exclusion criteria to specify ‘household

contact with children aged less than 2 years'. In practice, discharge from the quarantine facility requires resolution of diarrhoea and strict adherence to infection control practices to mitigate risk of onward transmission to all household and close contacts. If participants live with young children aged older than 2 years, participation would be based on an individualised discussion and consideration of risks addressing individual circumstances.

Reviewer: 2 (Dr. Helton C Santiago, Universidade Federal de Minas Gerais)

The manuscript bmjopen-2023-076477, by Smith et al., describes the plans of CHANTS study (Challenge Nontyphoidal Salmonella – CHANTS). This is the plan to develop a CHIM for NTS aiming to

gain insights into its physiopathology and to develop a tool for vaccine testing. Nontyphoidal salmonellosis is a severe disease with clinical presentation ranging from asymptomatic to sepsis with a high mortality rate. The development of the CHIM is well justified because of the social impact of NTS infection. In addition, the model will serve not only to accelerate an urgently needed vaccine, but also help the scientific community to understand the disease progression and presentation, perhaps allowing the identification of resistance surrogate markers.

The protocol is well described, and it is clear a lot of effort has been employed to identify and mitigate risks to guarantee the safety of study participant, to plan accurate decision taking along the protocol, and to promote accurate and relevant data collection. The authors will use a novel approach to escalate/de-escalate pathogen dosing, based on real-time data analysis from all the doses used (CRM

analysis). The aims are very clear and allow the best protocol progression to fulfil it. The protocol has received ethical approval and is expected to contribute significantly to the specific field of research.

There are no results presented as the manuscript is a protocol description.

We thank the reviewer for their thoughtful comments and positive feedback.

Additional Edits

- Exploratory objectives have been moved from the main body of the manuscript to supplementary table 1 to comply with the 2-page table limit as advised by the editorial office.
- The eligibility criteria table (Supplementary Table 2) and sampling time points and volumes (Supplementary Table 4) have been updated to reflect protocol amendments made since the original submission of this manuscript. Namely, addition of HLA-B*27 antigen positivity as an exclusion criterion, and clarification that only a recent history of malaria infection within the preceding 12 months precludes enrolment. All of these changes have been approved by the NHS London – Fulham Research Ethics Committee 21/PR/0051; IRAS Project ID 301659.